# Alternative Practices in Organic Dairy Production and Effects on Animal Behavior, Health, and Welfare

**DOI:** 10.3390/ani12141785

**Published:** 2022-07-12

**Authors:** Hannah N. Phillips, Bradley J. Heins

**Affiliations:** 1Department of Animal Science, University of Minnesota, Saint Paul, MN 55108, USA; 2West Central Research and Outreach Center, University of Minnesota, 46352 MN-329, Morris, MN 56267, USA

**Keywords:** organic, behavior, disbudding, human–animal relationship

## Abstract

**Simple Summary:**

The basis of livestock farming is preventing disease and improving animal welfare and well-being. Organic dairy farmers have very few options for the treatment of diseases and for the mitigation of pain in dairy calves and cows. Calving may be stressful for first-lactation cows because they must adapt to many different situations when they are milking. Alternative therapies to improve animal welfare must be researched in organic livestock production to verify that their use improves animal well-being. This review provides a brief background on organic production systems, illustrates current understanding of pain management for disbudding dairy calves, and discusses managing transition heifer behaviors and udder health to improve organic livestock well-being.

**Abstract:**

The number of organic dairy farms has increased because of the increased growth of the organic market, higher organic milk price, and because some consumers prefer to purchase products from less intensive production systems. Best management practices are expected from organic dairy farms to ensure animal health and milk production. Organic dairy producers typically transition from conventional systems to avoid chemicals and pesticides, enhance economic viability, improve the environment, and increase soil fertility. Organic dairy producers respect and promote a natural environment for their animals, is also an important component of animal welfare. Organic producers have few options to mitigate pain in dairy calves. In the United States, therapies to mitigate pain for disbudded organic dairy calves are regulated by the US National Organic Program. Organic producers regularly use naturally derived alternatives for the treatment of health disorders of dairy calves, heifers, and cows. Alternative natural products may provide an option to mitigate pain in organic dairy calves. Despite the reluctance to implement pain alleviation methods, some organic farmers have expressed interest in or currently implement plant-based alternatives. Efficacy studies of alternative remedies for organic livestock are needed to verify that their use improves animal welfare. Non-effective practices represent a major challenge for organic dairy animal welfare. The relationship between humans and animals may be jeopardized during milking because first-lactation cows may exhibit adverse behaviors during the milking process, such as kicking and stomping. The periparturient period is particularly challenging for first-lactation cows. Adverse behaviors may jeopardize animal welfare and reduce safety for humans because stressed heifers may kick off the milking unit, kick at milkers, and display other unwanted behaviors in the milking parlor. This may reduce milking efficiency, overall production, and ultimately reduce the profitability of the dairy farm. Positive animal welfare is a challenging balancing act between the three overlapping ethic concerns. Identifying animal welfare deficits in organic livestock production is the first step in capitalizing on these opportunities to improve welfare.

## 1. Introduction

### 1.1. Organic Livestock Production

The history of organic agriculture provides insight to the core values of today’s organic livestock industry. Agriculture polarized in the United States at the turn of the Environmental Revolution in the 1970s over concerns about chemical fertilizers and pesticides [1]. After years of organic industry groups requesting the protection of their farming practices, the US congress passed the Organic Foods Production Act of 1990, which created national standards for all aspects of organic agriculture to help unify organic practices. In 2001, the USDA created the National Organic Program (NOP) and Code of Federal Regulations (Title 7, Subtitle B, Chapter I, Subchapter M, Part 205) [2] to protect the integrity of the organic seal and mandate regulations. For example, all organic farms must undergo a certifying process by an NOP-accredited agency. Although there are several technical differences between organic and conventional livestock systems, the major defining characteristics include grazing and outdoor access requirements and the prohibition of most synthetic substances (e.g., antibiotics). The term “conventional” is an ambiguous term used to describe non-organic systems and—more than likely—intensive farming systems. However, there are some cases where these conventional, intensive, and non-organic farms may adopt some organic practices, such as grazing and alternative therapies. Henceforth, conventional is defined as “non-organic livestock systems that keep animals in total indoor confinement and have the ability to utilize treatments that are not allowed in organic practices, such as antibiotics, when necessary. Organic production systems are defined by the NOP as systems that are managed in accordance with the rules and regulations to respond to site-specific conditions by integrating cultural, biological, and mechanical practices that foster cycling of resources, promote ecological balance, and conserve biodiversity [2].

The National Organic Standards Board (NOSB), an advisory board for the NOP, reviews standards and reports recommendations to the NOP. For example, the NOSB may review and recommend the allowance of certain synthetic substances if a justified need exists and evidence supports their safety to people and the environment. If the NOP accepts the NOSB recommendations, the NOP initiates rulemaking to change The National List of Allowed and Prohibited Substances (§205.607) in the Code of Federal Regulations, which is available to the public [2]. The primary values of organic agriculture still exist in the modern organic livestock industry, and they serve as a foundation to support contemporary goals.

For health care, organic dairy producers should establish livestock health practices that focus on the prevention of disease and sickness. However, if management practices are inadequate to prevent illness, a producer may administer synthetic medications that are allowed under the NOP National List. Livestock producers should not withhold treatment from a sick animal to preserve the organic status of the animal. When methods of treatment in organic production fail, all methods must be used to restore an animal to health [2].

The organic industry is a fast-growing agricultural segment [3]. In the US, the organic livestock sector is dominated by dairy and poultry [4]. The top reported reasons why organic dairy producers choose to transition from conventional systems are to (1) avoid chemicals and pesticides, (2) enhance economic viability, and (3) improve the environment and soil [5]. These explanations expose modern motivations, yet reported themes still honor the earliest organic values of fostering natural systems.

### 1.2. Animal Welfare

Animal welfare is multifactorial; all components of an animal’s life contribute to its overall well-being [6]. There are several definitions of animal welfare, such as Broom’s 1986 definition, “The welfare of an individual is its state as regards its attempts to cope with its environment” [7]; The Five Freedoms developed between 1965 and 1979 [8,9]; and The Allostasis Concept, which appeared in 2007 [10]. Although these definitions contribute to the knowledge of animal welfare, the Fraser et al. [11] framework best describes how the organic industry values animal welfare.

In 1997, Fraser et al. [11] developed a holistic framework consisting of three overlapping ethical concerns in which animal welfare can be evaluated and human preference can be categorized (Figure 1). The framework’s ethical views are: (1) animals should be sound in terms of health and physiology (i.e., biological function), (2) animals should experience natural lives (i.e., natural living), and (3) animals should be free of negative emotional states (i.e., affective state). When evaluating animal welfare, people tend to emphasize the importance of one category over the others. For example, the NOP dairy standards value systems that mimic nature and commend practices that maximize the natural lives of animals—the natural living component of the animal welfare framework. Thus, organic producers tend to value natural living more than biological functioning and affective state when considering animal welfare [12].

Organic standards emphasize that animals should live according to nature, which may be accomplished by allowing animals to be reared with access to the outdoors, restricted periods of indoor confinement, and reduced stocking densities [13]. Animals raised organically may have more freedom to express natural behaviors compared to animals living in conventional systems. Furthermore, access to the outdoors may have an advantageous effect on animal health in some cases. In a review of literature on behavioral differences between cows housed with and without pasture access, Charlton and Rutter [14] suggested that the pasture environment may alleviate some animal health issues that are aggravated in total indoor confinement systems, such as lameness and hock lesions possibly caused by exposure to hard (e.g., concrete) flooring and resting areas. Alternatively, the pasture environment can introduce other challenges that may jeopardize animal welfare, such as biting flies [15,16], heat stress [17,18], an increase in gastrointestinal parasites [19], and impairment of the human–animal relationship [20,21] in dairy cows.

Animals living in organic systems may have some advantages for improved animal welfare compared to those raised in conventional systems, especially in terms of abilities to perform natural behaviors and alleviate animal welfare issues exacerbated by total confinement. However, the pasture environment presents its own animal welfare challenges, and there are several other facets of organic practices to deliberate upon that potentially affect animal welfare.

Placing most of the focus into the natural living component of animal welfare may be problematic for organic animals because emphasis in only one category comes at the expense of the others. To support this idea, previous literature acknowledged the deficits in organic livestock production regarding the biological function and affective state categories [22]. Bergman et al. [23] reported that organic dairy farms were less compliant compared to their conventional counterparts on the use of pain relief for disbudding calves, which may be partially due to the limited organic-approved options for pain relief. In a survey of veterinarian perspectives of organic livestock production, Sorge et al. [24] found that many veterinarians disagreed that animal health was improved on organic farms and expressed concern for the absence of proven therapies that may impair animal welfare on farms. Furthermore, veterinarians reported they struggled to successfully treat sick animals with alternative management practices within NOP guidelines [24]. It is evident that there are many disadvantages to organic animal production systems, especially when animals require a treatment intervention and alternative therapies fail.

It is noteworthy to acknowledge that animals have preferences within their living environment. Previous studies found that dairy cows prefer pasture, which is contingent on many factors, including time of day, weather, and individual variation [14,25,26]. It seems intuitive to think that animals raised in organic systems—where the freedom of choice is allowed—have better welfare, though the opportunity for choice may not necessarily relate to improved animal welfare, as animals may not choose what is in the best interest of their welfare.

Motivation tests have been used to determine the intensity of an animal who is willing to work to acquire a resource [27]. It has been suggested that strong motivation for a resource indicates that the resource is vital to the animal and denying that resource has a negative effect on animal welfare [28]. In an experiment by von Keyserlingk et al. [29], trained dairy cows pushed open a gate to access fresh feed or pasture. Cows pushed a similar weight to acquire feed and pasture but pushed more weight to gain pasture access at night [29]. Another experiment by Charlton et al. [26] found that dairy cows’ time on pasture declined when walking distance increased during the day but walking distance did not affect nighttime pasture use. Therefore, access to pasture may be an especially important resource for dairy cows at night. Therefore, the ability of an animal to access a resource that is highly important may influence animal welfare, but further research is required to verify whether having this access directly improves animal welfare.

There is currently no strong evidence on whether animals reared in organic systems have inferior or superior welfare compared with animals raised in conventional systems [30]. Furthermore, the level of animal welfare is likely contingent on various management factors and complex situations. For example, Sutherland et al. [31] reported that mastitis is a common and important welfare issue for dairy cattle regardless of organic status. While mastitis may be less common on some organic dairy farms [32], antibiotics are prohibited in organic production, so the ability to effectively treat organic cows for mastitis is limited. Ruegg [33] reported that alternative therapies—such as whey-based therapies, garlic tincture, and aloe vera—are commonly used to treat mastitis in organic cows, but limited research exists on whether these therapies are effective, and their use may actually prolong suffering. Positive animal welfare is a challenging balancing act between the three overlapping ethic concerns. Identifying animal welfare deficits in organic livestock production is the first step in capitalizing on these opportunities to improve welfare.

## 2. Pain Management for Disbudding

### 2.1. Horn Removal

Whether performed under conventional or organic management, horn removal is a major concern among the industry and the public [34,35]. However, the majority of dairy farms in the US (94%) remove horns [36]. Horns are perceived as a risk for animal and human injury and are therefore undesirable [37]. However, very little evidence has shown that horns are a risk for injury if farmers provide excellent housing and management and maintain a suitable human–cattle relationship [38]. Moreover, horn removal may have little benefit to animal and human safety [39]. At the present, there is evidence of stakeholder interest in preserving horns [34,40]. In the US, there are no current studies on horned dairy cattle, so it is difficult to accurately enumerate the presence of horned organic herds. In the European Union, a survey of 419 dairy farms estimated that 78% of organic farms had animals with horns [41]. Perhaps unaccounted horned organic dairy herds exist in the US, especially considering current trends in the European Union. Preserving horns as a strategy to enhance dairy cattle welfare is insufficiently investigated and represents a research topic of high priority. However, horn removal is still dominant in the organic dairy sector [5,23]; thus, scientific investigation on ways to mitigate pain inflicted by horn removal procedures still demands continuation. Despite a reluctance to implement pain alleviation methods, organic dairy farmers support disbudding as an accepted practice. However, organic dairy farmers are exploring other alternatives to disbudding, such as polled genetics [42].

Dehorning is the most painful and least desired method of horn removal [43] and is defined as “The process of removing the horn of an adult cow after the horn has developed attachment to the skull” [43]. Therefore, the dairy industry has advocated for farmers to disbud calves instead [44]. Disbudding is defined as “the process of damaging the horn bud in young calves to prevent the growth of horns” [43]. Over the years, disbudding has increased in popularity as a method of horn removal, such that disbudding was implemented on 86% of dairy farms in 2014 the US. The two major methods used to disbud calves include cauterization and caustic paste [36]; however, caustic paste is generally not approved for organic use, since it contains chemicals that destroy the horn bud tissue after topical application (§205.603). Furthermore, the use of caustic paste can be problematic, since it has been demonstrated in clinical trials to cause pain and become dangerous if accidently transmitted to other body parts [45,46]. Therefore, caustic paste should be promoted with caution, since it could encourage farmers to rear calves in isolation, which has detrimental effects on animal welfare [47]. Therefore, cautery disbudding represents the primary method of horn removal in organic dairy calves and a widespread animal welfare issue for the organic sector.

Pain is the most significant acute effect of cautery disbudding. Calves exhibit intense and frequent escape behaviors during disbudding [48] and elevated pain and wound sensitivities for at least 24 h after the disbudding [49,50]. Stewart et al. [51] showed deviations in ocular temperature within minutes after disbudding, suggesting immediate pain following disbudding. Neave et al. [52] found that calves were less likely to complete an ambiguous task at 6 and 22 h after disbudding, suggesting “pessimism” in disbudded calves. Recent studies even suggest that disbudded calves experience prolonged pain before [53] and after [54] the wounds re-epithelialize, which takes approximately 9 weeks [53]. The long-term pain of disbudding is poorly understood and could have ramifications on the welfare of adult cows.

Therefore, disbudding is a major animal welfare concern with potential long-term negative effects, and strategies to minimize pain should be utilized. The NOP recommends instilling practices which minimize acute pain and stress caused by the disbudding procedure using effective methods and approved therapies. However, organic producers have limited pain mitigation therapy options (§205.238) [2], making disbudding pain management a challenge and widespread animal welfare issue for the organic sector.

### 2.2. Pain Management

The best way to alleviate acute disbudding pain is through multimodal therapy—using multiple methods to manage pain. In a review of 21 studies by Winder et al. [55], it was suggested that the combination of a cornual nerve block with an anesthetic and a systemic non-steroidal anti-inflammatory drug (NSAID) increases acute numbness compared to a local anesthetic alone. A local anesthetic induces numbness in the horn bud area, and the NSAID systemically reduces inflammation by inhibiting the enzyme cyclooxygenase (COX) and consequent synthesis of inflammatory prostaglandins, such as prostaglandin E2 (PGE2; [56]). This multimodal method is useful because local anesthetics have a functional duration of approximately 90 min [57], and a long-lasting NSAID may alleviate the inflammatory pain thereafter [55]. However, multimodal pain mitigation therapies are rarely implemented on organic dairy farms.

Pain alleviation methods for disbudding are quite low and depend on several factors of feasibility. A recent survey of 189 US organic dairy producers reported that less than 26% of farms used either a local anesthetic or an NSAID for disbudding calves [23], and the use of multimodal pain therapy is estimated to be rare [58]. Organic producers are restricted to substances that are approved by the NOP (§205.603), and the few NSAID options available limit the feasibility of proper pain alleviation. For example, lidocaine (e.g., local anesthetic) and aspirin (e.g., NSAID) were added to the NOP National List of substances in 1995 and are generally acknowledged as substances that accommodate organic values [59]. However, aspirin is not approved by the Food and Drug Administration (FDA) for use in cattle and is therefore not allowed. In 2007, flunixin (e.g., NSAID) was added to The National List of Allowed and Prohibited Substances in light of its positive impact on animal welfare [59]. However, flunixin was simultaneously strongly opposed by farmers and NOSB reviewers, who were charged by its contradiction of organic values [59]. Furthermore, flunixin must be administered intravenously (i.v.), which may be a contributing factor to its lack of adoption, since i.v. methods may be challenging and unappealing to some producers [60]. Consequently, organic farmers have demonstrated reluctance to implement flunixin as a post-operative pain management therapy but have expressed interest in plant-based alternatives to alleviate pain [32]. Furthermore, xylazine is allowed for use under the USDA-NOP but must be used by or under the direction of a veterinarian (The National List of Allowed and Prohibited Substances (§205.603) [59]. In Finland, Adam et al. [61] reported that a low dose of xylazine allowed for sufficient sedation as a local anesthetic for disbudding in Finnish Ayrshire calves. However, xylazine does have a side effect of decreasing core body temperature after injection for dairy calves that were disbudded [62]. Vickers et al. [45] recommend that xylazine should be used when disbudding with caustic paste, even though xylazine does not have an anesthetic effect. Recently, calves sedated with xylazine prior to disbudding had less response to pain stimuli and greater rates of play behavior following sedation [63].

Lidocaine as a local anesthetic is approved as a cornual nerve block in organic dairy cattle. However, lidocaine use requires a withdrawal period of 6 days after administration to dairy calves that are disbudded [59]. Lidocaine 2% is a commonly used synthetic substance for organic livestock and alleviates disbudding pain by providing local analgesia [55]. Lidocaine provides analgesia the horn bud area within 2 to 5 min and has a duration of 90 min. Organic dairy producer and veterinary stakeholders have either adopted or exhibited an interest in non-synthetic substances, such as herbal therapies, to mitigate disbudding pain [23,32]. A survey of over 180 US organic dairy farms reported that a quarter of dairy farms used natural therapy as pain management for horn removal procedures [23]. However, these alternative therapies may be a problematic solution, since their efficacy is mostly based on anecdotal evidence. A survey of over 150 US organic dairy producers found reduced knowledge of farmers about effective organic-approved practices [64]. Furthermore, alternative practices have been identified as a major threat to organic dairy welfare [65]. Recently, Barkema et al. [66] proposed that future research should identify organic-approved alternative remedies that are effective for reducing pain.

Pain and stress are challenging to quantify and understand in animals. Physiological measures of pain can be useful but also require careful interpretation [49]. The body responds to pain by triggering an autonomic nervous system (ANS) response [67]. In particular, the sympathetic nervous system (SNS) of the ANS orchestrates a fight-or-flight response, in which the brain communicates to the adrenal gland via converging systems; the sympathetic–adrenal–medullary (SAM) system uses electrical signals, and the hypothalamic–pituitary–adrenal (HPA) axis uses a series of cascading hormones to prompt the adrenal gland [67]. The SAM system quickly triggers the adrenal gland to release catecholamines, such as adrenaline and norepinephrine, to increase vigilance and ultimately prepare the body for immediate physical reaction [67]. The HPA stimulates the adrenal gland to release cortisol, which may have a variety of prolonged functions, including immune and inflammatory suppression [68]. Therefore, pain and stress in animals can be inferred by observing elevated hormones involved in the SAM and HPA axis function [68]. However, the HPA axis hormones may be problematic measurements of pain since they elevate in response to other categories of stressors.

Quantifying pain-specific behaviors that increase in frequency after disbudding (e.g., ear flicks and head rubs) is another useful tool to draw conclusions about pain in disbudded calves [69]). However, as behavior measures may be inconsistent between studies, subjective, time-consuming, and variable between individual animals [55,60], it is important to examine diverse pain characteristics in examinations of disbudding pain in calves.

### 2.3. Alternative Non-Steroidal Anti-Inflammatory Drugs—Synthetic Salicylates

Synthetic salicylates, such as acetylsalicylic acid (i.e., aspirin) and sodium salicylate, have previously been used as effective anti-inflammatories, antipyretics, and analgesics in cattle. In an experiment by Coetzee et al. [70], intravenous sodium salicylate administered at a dose of 50 mg/kg reduced cortisol concentrations when compared to untreated cattle following castration. However, a 50 mg/kg oral dose of aspirin did not mitigate the cortisol response Coetzee, et al. [70]. In another experiment, Baldridge et al. [71] reported that sodium salicylate dissolved in ab libitum water at rates of 2.5 to 5.0 mg/mL and offered from 1 day before to 2 days after castration and dehorning improved ADG for the next 13 days and decreased cortisol concentrations for up to 6 h after the procedures compared to calves that received no treatment. Although synthetic salicylates show promising utility for pain mitigation in cattle, they have never been officially approved by the FDA and are therefore not permitted.

### 2.4. Alternative Non-Steroidal Anti-Inflammatory Drugs—White Willow Bark

White willow (*Salix alba* L.) bark (WWB) is one of the most popular plant-based therapies used for pain relief [72]. As with all plants from the Salix genus, white willow bark contains salicylate compounds primarily comprised of salicin [73], which is converted into salicylic acid (SA) when consumed orally [74]. Salicylic acid is similar to synthetic salicylates, such that it inhibits the enzyme COX and blocks inflammatory prostaglandins, such as PGE2 [75]. Various studies reported reductions in pain when administering WWB to humans [76,77].

White willow bark may be a useful alternative to synthetic salicylates to mitigate the delayed onset of pain in disbudded calves. Plant matter, especially leaf and branch trimmings, from the Salix genus have been previously demonstrated to be a nutritious feed source in agroforestry systems and safe for consumption by ruminants [78,79,80,81], but the efficacy of WWB as an alternative therapy to alleviate pain in cattle is currently unsupported by scientific evidence. Furthermore, animal welfare critics of the organic dairy industry constantly reference unproven alternative therapies as a major animal welfare concern [12,22,23]. Therefore, it is essential that scientific research begins filling this exposed knowledge gap by investigating WWB for its analgesic effects in calves. Recently, Phillips et al. [42] reported that white willow bark contains 0.22% salicin. For blood plasma concentrations of the inflammatory biomarker PGE2, flunixin meglumine lowered PGE2, whereas white willow bark was ineffective at reducing PGE2 and achieving the minimum salicylic acid concentration necessary for analgesia in calves. The results indicated that white willow bark provided in three oral doses was unsuitable for producing analgesia in calves [42].

Salicin is the most prominent compound in WWB extracts that is responsible for anti-inflammatory effects [82]. However, the amount of salicin in WWB products is not commonly provided by manufacturers. In an observational study to evaluate the amount of salicin in the bark of various Salix species grown in Lithuania, Kenstavičiene et al. [83] found that WWB contains 1.21 to 1.87% salicin. Furthermore, Pitta et al. [79] and McWilliam et al. [80] reported that leaf and branch trimmings from Salix species contained 0.09 to 0.17% salicin. High-performance liquid chromatography (HPLC) is the most common method of determining the concentration of salicin in plant matter. The amount of salicin in WWB products, such as ground and dried WWB powder, is not typically evaluated by manufacturers. Therefore, the salicin concentration of several WWB products that are currently used or may be used by the organic dairy industry to mitigate pain will be evaluated using HPLC.

After ingestion, salicin is converted to several different metabolites from the salicylate family which can be detected in the plasma of blood. Salicylic acid is the major metabolite that makes up total salicylates detected in the plasma after ingesting salicin. In a pharmacokinetic experiment of oral WWB in humans, salicylic acid made up 86% of the total detected salicylates in the blood serum [84]. The minimum total salicylate plasma concentration needed for analgesia in calves was previously estimated to be 25 to 30 μg/mL [85]. Since SA makes up an estimated 86% of total salicylates in the plasma after ingesting salicin [84], the estimated minimum SA plasma concentration needed for analgesia in calves is approximately 21.5 to 25.8 μg/mL. Therefore, plasma concentrations of SA will be measured in calves receiving WWB to determine if the minimum SA plasma concentration needed for analgesia in calves is met and to corroborate inflammation findings.

Non-steroidal compounds prevent inflammation by inhibiting COX, the class of enzymes involved in the production of inflammatory prostaglandins [86]. Prostaglandin E2 is the most notable inflammatory prostaglandin because of its superior effect on the processing of pain signals [87]. COX-1 and COX-2 are the two types of COX enzymes. Prostaglandins related to COX-1 control homeostatic processes and are involved in the resolution of inflammation, but not the progression of inflammation [88]. Prostaglandins related to COX-2 are associated with inflammation from tissue injury [88]. Few studies investigate the specific mechanisms of WWB on COX enzymes. In one study [89], white willow bark inhibited COX-2-mediated PGE2 release in vitro. In an investigation of aspirin and salicylate, which have similar mechanisms to salicin, Higgs et al. [90] showed that both NSAIDS mediated PGE2. Furthermore, prostaglandin E2 has been commonly used as a measurement of inflammation in cattle [91,92]. Therefore, prostaglandin E2 will be measured in calves to understand the effects of WWB on inflammation.

## 3. Managing Transition Organic Dairy Heifer Behaviors and Udder Health

### 3.1. Challenges of Mastitis for Organic Dairy Farms

The National Organic Program of USDA sets the standards to which organic farmers have to adhere in order to produce organic products [2]. Organic dairy farming focuses on disease prevention and limits the use of synthetic drugs for the treatment of livestock diseases. For example, antibiotics are not allowed to treat animals unless the animals leave organic production immediately after. Unfortunately, some animals will still become sick despite best preventative practices.

In dairy cows, mastitis is one of the most common and economically important dis-eases [93]. Mastitis is an inflammation of the udder and will affect not only the animal’s well-being, but also the milk’s quality. In conventional production, mastitis is most commonly treated with intramammary antibiotics. However, this is not allowed for organic systems, and effective alternative treatment approaches are needed [32].

Udder health is important for the sustainability and optimal productivity of a dairy farm [94]). Milk from healthy cows, reflected by a low somatic cell count, has an improved shelf life and therefore receives a premium price. In addition, international trading partners such as Europe require on-farm bulk tanks with SCC under 400,000 and standard plate bacterial counts of less than 100,000 colony-forming units. Tikofsky et al. [95] reported that SCC for organic farms in New York averaged 273,000, whereas Zwald et al. [96] reported that 47% of organic farms in the upper Midwest had SCC greater than 300,000 and 15% had SCC greater than 400,000. Unfortunately, mastitis remains a common disease on dairy farms and is a leading cause for culling of cows [97]. The disease can reduce milk production and milk quality, impair animal welfare, and increase veterinary and labor costs. Effective treatment options beyond antibiotics are lacking [98]. Therefore, it is crucial for organic dairy producers to use effective strategies to prevent this disease and its associated losses. Recently, Hardie et al. [93] reported a mastitis incidence (13.8%) from organic Holsteins cows in the US and Ahlman et al. [97] reported that poor udder health is the main reason for culling cows in organic herds. Recently, Fernandes [99] reported that elevated SCC in the first month of lactation had detrimental effects on the milk yield and survivability of dairy cows in USDA organic herds.

Organic dairy farms have reported some success and failures [30] with using alternative products for mastitis in cows. However, farmers have reported drying off the affected quarter, or—in severe cases—culling the animal as opposed to using alternatives to antimicrobials [100]. Mullen [101] evaluated the pharmacokinetics of garlic, thymol and carvacrol for use in controlling *S. uberis*-induced mastitis and reported that withhold times of at least 24 h should be established in organic herds that use these products. However, these products did not produce bacterial cures for mastitis [101]. Furthermore, researchers reported the efficacy of the herbal products (Phyto-Mast and Cinnatube) was similar to conventional therapies, and the products did not have any adverse effects on cows [102] Frequent stripping or the use of a topical udder rub are commonly used on organic farms [5]. The idea behind frequent stripping is that it removes the bacteria and bacterial toxin load from the udder to improve healing. Similarly, topical udder creams with peppermint or similar components are thought to decrease swelling and to improve blood flow and thereby improve the clearance of an infection from the udder. Although the rationale of both approaches is plausible, there are few data supporting the use of either therapy approach as effective treatment of clinical mastitis.

In dry-off, milk production is stopped, and in conventional and intensive dairy systems, therapeutic intervention is provided to cows to clear existing infections. However, intramammary antibiotics in dry-off are not allowed under the USDA National Organic Standards [2]. Some organic dairies may administer a variety of nonantimicrobial organic products [30,32], but clinical efficacy is lacking [33].

The dry period provides the udder with important time to regenerate and prepare for the next lactation. However, during the dry period, cows may be vulnerable to intra-mammary infections that may persist through the dry period and subsequently cause clinical mastitis early in lactation. Currently, in conventional and intensive systems, the dry-off procedure includes abrupt cessation of milking and applying blanket antibacterial treatment to prevent early dry period infections, but antibiotics are not allowed for organic herds. Current thought is to lower milk yield at dry-off to help prevent new infections during the early dry period [103], but reduction in feed has been associated with increased stress and metabolic disease incidence in dairy cows [104]. Intermittent milking at dry-off may reduce milk production with little to no discomfort to cows [104].

### 3.2. Challenges for Early-Lactation Heifers

First-calf heifers encounter several challenges following calving that can jeopardize animal welfare. Firstly, some heifers may become distressed when they encounter unfamiliar experiences related to being milked, such as unfamiliar sounds and smells in the milking parlor and tactile stimulation to the udder by handlers and milking units. Van Reenen et al. [105] reported that peak plasma cortisol concentrations were approximately 20% greater for heifers during milking on day 2 compared to day 130 of lactation, indicating that the beginning of the lactation period can be stressful. The typical lactation period is approximately 305 days, so 130 DIM represents mid-lactation. Sutherland and Huddart [31] also found similar results, in which heifers had 2.0 times the plasma cortisol concentration on the first DIM compared to the fifth DIM. Furthermore, authors also reported that plasma oxytocin concentrations after milking preparation procedures (but before milking unit attachment) were 2.4 times greater for heifers at 130 DIM compared to 2 DIM, indicating that heifers may need time to acclimate to milking [105]. Oxytocin is defined by the National Mastitis Council (https://www.nmconline.org/ (accessed on 9 November 2021)) as “the hormone produced in the pituitary gland that causes milk let-down”.

Distressed heifers can endanger human handlers, because heifers may kick off milking clusters, kick at milkers or display undesirable behaviors that interfere with milking. This may increase injury to milkers and increase the risk of mastitis for the heifer [31,105]. However, many dairy farms already have voluntary milking systems and with these systems the risk of injury to the milkers is reduced or eliminated. Mastitis is defined by the National Mastitis Council (https://www.nmconline.org/ (accessed on 9 November 2021)) as “inflammation of the udder, most commonly caused by an infecting microorganism”. For example, a prospective evaluation of all injuries by cattle at a hospital in New Zealand over a 1-year period conducted by Watts and Meisel [106] showed that hand or wrist injuries were commonplace and occurred after being kicked by a cow at milking time. In terms of udder health, Nitz et al. [107] found that heifers that detached milking cups during milking were 2.6 times more likely to develop new intramammary infections (IMI) between 3 and 17 DIM. In a study of 46 farms in Switzerland, Ivemeyer et al. [108] found that the number of kicks per cow displayed during milking was associated with new IMI infection incidences. Intramammary infection is defined as “the presence of an organism in the udder that is isolated from a milk sample”. Therefore, aversive heifer behaviors during the early-lactation period may jeopardize both human and animal welfare.

In general, heifers are vulnerable to clinical mastitis and IMI during early lactation [108,109,110,111]. Clinical mastitis is defined by the National Mastitis Council (https://www.nmconline.org/ (accessed on 9 November 2021)) as “udder inflammation characterized by visible abnormalities in the udder or milk”. In an observational study of 1014 heifers in Sweden, Persson Waller et al. [111] reported that 50% of the 364 recorded mastitis cases in heifers occurred within the first 6 DIM, and were primarily diagnosed as Staphylococcus aureus. This is a concern for farmers since poor udder health in heifers is associated with production, treatment, and labor costs. In 2009, Huijps et al. [112] estimated that the costs of clinical mastitis and IMI were $18.75 and $6.56 per heifer, respectively. In a more recent study in 2014, Cha et al. [113] estimated that the average cost of a clinical mastitis case ranged between $115 and $476 after considering mortality and reduced conception costs. Furthermore, poor udder health in early lactation may also put heifers at risk of future infections [114]. Poor milking behavior may increase the economic loss for farms due to increased risk of IMI [107], decreased milk productivity [115], and the risk of early culling [116]. The main reason for the culling of organic first-lactation cows was mastitis [97]. Furthermore, lower somatic cell score is associated with improved longevity of organic cows, because lower somatic cell score is associated with reduced incidences of mastitis [117]. Culling is the main management strategy for reducing mastitis in organic dairy herds, and heifers with mastitis during their first lactation were more likely to be culled than those heifers without mastitis. Rearing of organic dairy heifers is very costly because of high feed costs [99] and therefore, it is imperative to reduce mastitis in heifers.

### 3.3. Methods to Modulate Aversive Behaviors and Mastitis

Several approaches have been considered to reduce distress and prevent mastitis in heifers. In general, these strategies include handling heifers and familiarizing them with the milking parlor before calving [118,119]. For example, Hemsworth et al. [116] found that heifers that were accustomed to handling during calving had 40% fewer flinch, step, and kick responses during milking during the first 20 DIM compared to heifers that were not handled during calving. Bertenshaw et al. [120] reported that brushing heifers for 30 to 245 min during the last 6 weeks of gestation reduced kicking during milking up to the first 28 DIM compared to heifers that were not brushed. Das and Das [121] found that 30 udder massage sessions lasting 20 min each during the last 2 months of gestation improved temperament, milk letdown and milk flow rates over the first 16 DIM. Eicher et al. [118] reported that heifers that moved through the parlor (but were not milked) with lactating cows twice daily for 3 weeks prior to calving balked for a shorter amount of time while entering milking stalls on the first DIM compared to heifers prior to calving that did not receive any treatment. However, behaviors of shifting, stomping, kicking and kicking the milking unit off during milking were similar among treatments on the first DIM [118]. Kutzer et al. [119] reported that acclimation before calving, which consisted of familiarizing heifers to the milking herd 10 days prior to calving and moving them through the milking parlor on at least 10 visits, reduced post-parturient stepping, kicking, ear-flattening, tail-tucking, and eye-widening behaviors in heifers over the first 7 DIM. However, intensive protocols to acclimate heifers to milking procedures may not be feasible for many farms due to labor challenges, so developing a protocol that fits within the capabilities of dairy farms is necessary.

A variety of strategies implemented during the pre-parturient period have been explored to prevent clinical mastitis and IMI, such as internal teat sealants [122], antibiotic therapies [123], milking [124], and repeated use of teat dip or spray [111]. However, some of these strategies, such as teat sealants and antibiotics, are not allowed in organic dairy animals in the US. In one experiment by Santos et al. [125], pre-parturient milking three times daily for 15 days prior to calving lowered the number of heifers with positive bacterial milk cultures by 25% on the first DIM and decreased the incidence of mastitis by 57% during the first 135 DIM. In another experiment, Lopez-Benavides et al. [126] reported that pre-parturient teat-spraying with an iodine-based disinfectant three times weekly for 21 days prior to calving reduced Streptococcus uberis in milk samples immediately after calving by 53% but did not reduce clinical mastitis. However, a reduced labor force may prevent the adoption of these strategies on many farms. Therefore, current pre-parturient strategies for preventing clinical mastitis and IMI in heifers must be improved to be practicable on farms in terms of labor limitations.

Aversive behaviors are behaviors that are undesirable to human handlers. These include behaviors that endanger handler safety and behaviors that interfere with milking efficiency. Commonly examined aversive milking behaviors include stomping, kicking, and kicking the milking unit off. Ease of milking parlor entry is also important, as aversive behaviors such as balking may interfere with milking efficacy [118]. Furthermore, objective temperament scores are commonly used to describe the overall reactivity of cows to stressors related to milking [121]. Aversive behaviors may also be indicative of distress in heifers. Temperament scores and measurements used in current assessments include milking speed, milk flow rate, approach test, novel test, handling temperament, heart rate, general temperament, and automated milking system temperament [127]. Hemsworth et al. [116] found that milk cortisol concentrations were associated with flinch, step, and kick responses in heifers, indicating that these behaviors may be indicative of distress. Fogsgaard et al. [128] reported cows with mastitis were more restless during milking, indicated by greater frequencies of tripping and kicking, suggesting that the presence of these behaviors may indicate pain caused by mastitis.

Furthermore, Phillips et al. [129] found that first-lactation cows that had their teats cleaned and were teat-dipped weekly 3 weeks prior to calving had reduced kicking and restlessness behaviors during post-calving milking. Cows had lower IMI caused by Staphylococcus aureus post-calving. Adjusting heifers to the milking parlor prior to calving may improve first-lactation cow well-being and promote a positive human–animal relationship.

## 4. Conclusions

Organic dairy production is a worthwhile method of dairy farming with steady and emerging markets. However, many farmers are apprehensive of organic dairy production practices because of concerns that no antibiotic use may have a negative impact on herd health. Alternative therapies to improve animal welfare must be researched in organic livestock production to verify that their use improves animal well-being. Critics of organic dairy management practices are concerned that producers use ineffective approaches to care for animals. However, the successful management of organic dairy herds depends on disease prevention through the use of traditional good husbandry practices.

## Figures and Tables

**Figure 1 animals-12-01785-f001:**
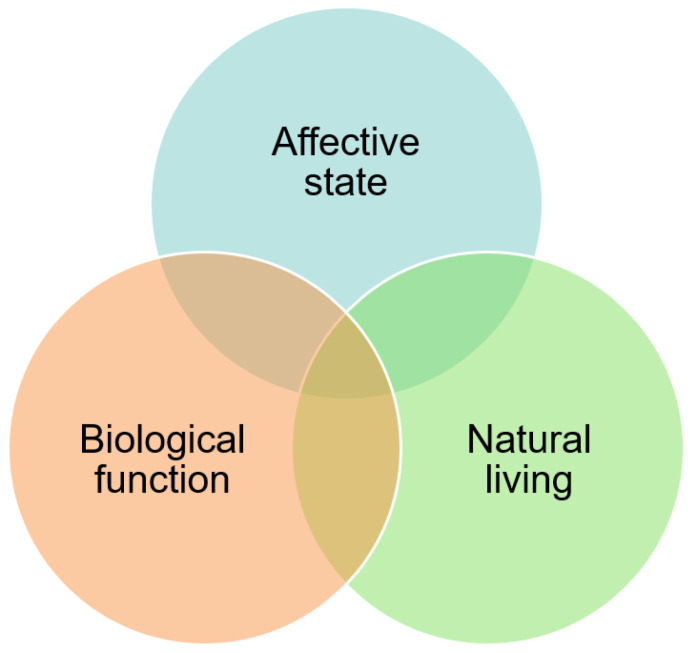
The animal welfare framework. Descriptions were developed by Fraser et al. [11].

## Data Availability

Not applicable.

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
