# Peer review of "Alternative Practices in Organic Dairy Production and Effects on Animal Behavior, Health, and Welfare"

_animals, 2022, doi:10.3390/ani12141785_

Round 1

Reviewer 1 Report

Alternative practices in organic dairy production and effects on animal behavior, health, and welfare

Thank you very much for allow me to read this interesting manuscript. I really
enjoyed reading it. The manuscript is well written and pleasant to read.
The manuscript is focused on discuss some management practices that can differ between organic and non-organic dairy operations.
I found interesting arguments about the features in organic dairy management
that may impact animal health or welfare. However, I was wondering why some specific points commonly discussed on organic dairy production were not mentioned, e.g., some management practices regarding udder health such as drying-off protocols, reproductive management and record management.

Please see below a few specific comments about the manuscript:

L 97:

 * "Organic Animal Welfare" seems a little weird...

L 191:

 * This may be a controversial statement.

L 228-230:

 * This sentence doesn't seem to be well integrated into the main argument of
   this paragraph.

L 238:

 * There are additional medications used in protocols to control pain when
   disbudding calves.
   The use of sedatives such as xylazine, or injectable anesthetics for instance.
   Do you think it would be valuable to include some references to the use of
   this type of medications?

L 396-398:

 * This sentence is more related to the information in the next paragraph.
   This information also seems to be redundant with the idea developed
   in the following sentences (L 399-413).

L 490-491:

 * I'm not sure if your literature review supports this statement.

Author Response

Reviewer #1 comments:

Thank you very much for allow me to read this interesting manuscript. I really enjoyed reading it. The manuscript is well written and pleasant to read. The manuscript is focused on discuss some management practices that can differ between organic and non-organic dairy operations. I found interesting arguments about the features in organic dairy management that may impact animal health or welfare. However, I was wondering why some specific points commonly discussed on organic dairy production were not mentioned, e.g., some management practices regarding udder health such as drying-off protocols, reproductive management and record management.

Response: Thank you for your great comments on our manuscript. We appreciate the time you took for a thorough review.  The manuscript has improved based on your comments.  We have added some new paragraphs on mastitis in organic farms, as well as dry off according to reviewers’ comments. Thank you.

Please see below a few specific comments about the manuscript:

L 97:  "Organic Animal Welfare" seems a little weird...
Response: We agree and have changed it to just Animal welfare

L 191:  This may be a controversial statement.

Response: We have added additional text to explain the situation and statement.

L 228-230: This sentence doesn't seem to be well integrated into the main argument of
   this paragraph.

Response: We agree that the sentence does not fit at the end of the paragraph on disbudding. We have removed this sentence.

L 238: There are additional medications used in protocols to control pain when
   disbudding calves.    The use of sedatives such as xylazine, or injectable anesthetics for instance.
   Do you think it would be valuable to include some references to the use of
   this type of medications?
Response: Thank you for this suggestion.  We have added some references on using xylazine prior to disbudding in calves.

L 396-398: This sentence is more related to the information in the next paragraph.
   This information also seems to be redundant with the idea developed
   in the following sentences (L 399-413).

Response: Thank you for catching this repetitive sentence and the thorough review.  We agree and we have removed this sentence at the end of this paragraph.

L 490-491: I'm not sure if your literature review supports this statement.

Response: We agree, and we have removed this statement.

Reviewer 2 Report

The article is well structured, well documented with relevant and current bibliography and addresses topics of utmost relevance for dairy cattle in organic production.

Author Response

The authors wish to thank the three reviewers again for their efforts for review of this manuscript.  We have accepted all of the suggestions for improvement and have made changes to the manuscript. The revised manuscript is much improved based on comments of reviewers and have clarified the random regression equations. Thank you. We appreciate your thoughtfulness.

Reviewer #2 comments;

Comments and Suggestions for Authors:

The article is well structured, well documented with relevant and current bibliography and addresses topics of utmost relevance for dairy cattle in organic production.

Response: Thank you for your great comments on our manuscript. We appreciate the time you took for a thorough review.  The manuscript has improved based on your comments. 

Minor Concerns:

Line 128 to 130: They should include other welfare problems associated with pasture, which are documented in the scientific literature, such as the incidence of parasitic diseases and impairment of the human-animal relationship.

Response: Thank you for the comment.  We have added some references that indicate an increase in GI parasites and impairment of the human-animal relationship. 

Line 177 to 182: It would be important to more specifically document the effectiveness of alternative therapies in preventing and treating mastitis.

Response: We have added new paragraphs in the mastitis section that also discusses some of the alternative therapies used in treating mastitis in organic herds.

Line 293 to 296: The statements should be clarified and possibly put first the adverse effects of cortisol and later the importance of its secretion by the animal. Because the way it is written, it is interpreted that the secretion of cortisol is advantageous for the animal.

Response: We have looked at these 2 statements and agree that they needed to be clarified.  However, upon further review, we feel that these statements are out of place in the current paragraph, and we have discussed these thoughts previously as well, so therefore, we have removed these sentences in this paragraph.

Line 396 to 398: It is convenient to explain and detail the reasons that can lead to an increase in the incidence of mastitis.

Response: Again, thank you for pointing out these sentences.  Based on comments of another reviewer, we have removed this sentence because it did not fit well with this paragraph.

Line 401: Bearing in mind that many dairy farms already have voluntary milking systems, it should be noted that in such cases the risk of injury to the milkers is eliminated.

Response: We agree.  We have added a sentence that reflects this thought based on wording provided. Thanks.

Line 427 to 429: The work would be enriched if a study on the problem of early culling heifers in organic production were mentioned, and also if a description of some strategies were made to prevent mastitis in heifers.

Response: We have added some sentences that reflect the problem with culling and mastitis in heifers. Thanks for the suggestion.

Line 473 to 475: It would be very important to describe the objective temperament scores most used in current assessments.

Response: We have included some scores and measurements that are in used currently based on references.

Reviewer 3 Report

Reviewer comments for manuscript ID animals-1772681 entitled ‘Alternative practices in organic dairy production and effects on animal behaviour, health, and welfare’

General Comments

It is an interesting review on the contemporary aspects of organic dairy production. Organic dairy farming is an alternative to check the rampant factory farming of livestock where animal welfare is severely compromised. However, the viability of this segment of farming is dependent on evidence based and scientifically proven alternative practices to meet consumer demands, aspirations, and confidence.There are many conflicts where replacement of current practices in conventional or factory farming might be counterproductive towards the health and welfare of livestock. It is a nice attempt by the authors to draw attention to this alternative farming practice that hold promise due to consumer and public demand.

The authors have mainly looked into disbudding, pain management and mastitis in context of organic animal husbandry. There are other vital aspects of organic farming that also need attention/ reference such as management of parasitism (ecto and endo), dry cow therapy, teat dipping, milking machines versus manual milking, weaning of calves versus calves at foot, vaccination of livestock, castration, artificial insemination versus natural service (theory propagated by animal welfare activists, flooring, ration formulation, use of probiotics, nutraceuticals. I would like the authors to discuss these aspects for a more composite approach to this husbandry and to make this manuscript more comprehensive.  

I have a question for the authors to discuss - Is disbudding an accepted practice in organic livestock production as it is a routine procedure in intensive systems where spacing between animals is kept to a minimum? Disbudding is not natural and has welfare effects. Please clarify.

The manuscript is well written and nicely presented. However, the conclusions are very superficial and need to provide a broad overview and a take home message. At this moment I would like to see work on suggestions provided in my general and specific comments, before I recommend the manuscript for publication.

Specific Comments

Line 10: Is this not the basis of every animal husbandry practice rather being exclusive for organic livestock farming? Please clarify or reframe.

Line 21: Please replace ‘should be determined’ with ‘are expected’

Lines 22-24: Incomplete sentence. Please complete it.

Lines 33-39: How is this different from the intensive livestock farming? These problems are not exclusive for organic livestock farming. Please clarify.

Lines 59-60: What about ‘intensive systems? Please clarify and comment.

Line 64: Please replace ‘unallowed’ with ‘not allowed’

Line 64-68: Please insert a reference to support this statement.

Line 81-84: Please insert a reference to support this statement.

Lines 89-90: I am sorry I am not able to understand the sentence. Please use a simpler language for the benefit of the reader.

Line 35: Please replace ‘consider’ with ‘deliberate upon’

Lines 381-430: These are challenges for the entire cattle farming industry whether organic, intensive, or conventional. Please discuss the dilemmas of organic farming in context of mastitis.

Lines 455-68: Please refer to studies on organic teat dips or organic dry cow therapies in this paragraph.

Line 488: Please replace ‘suspicious’ with ‘apprehensive’

Author Response

The authors wish to thank the three reviewers again for their efforts for review of this manuscript.  We have accepted all of the suggestions for improvement and have made changes to the manuscript. The revised manuscript is much improved based on comments of reviewers and have clarified the random regression equations. Thank you. We appreciate your thoughtfulness.

Reviewer #3 comments:

General Comments

It is an interesting review on the contemporary aspects of organic dairy production. Organic dairy farming is an alternative to check the rampant factory farming of livestock where animal welfare is severely compromised. However, the viability of this segment of farming is dependent on evidence based and scientifically proven alternative practices to meet consumer demands, aspirations, and confidence.There are many conflicts where replacement of current practices in conventional or factory farming might be counterproductive towards the health and welfare of livestock. It is a nice attempt by the authors to draw attention to this alternative farming practice that hold promise due to consumer and public demand.

The authors have mainly looked into disbudding, pain management and mastitis in context of organic animal husbandry. There are other vital aspects of organic farming that also need attention/ reference such as management of parasitism (ecto and endo), dry cow therapy, teat dipping, milking machines versus manual milking, weaning of calves versus calves at foot, vaccination of livestock, castration, artificial insemination versus natural service (theory propagated by animal welfare activists, flooring, ration formulation, use of probiotics, nutraceuticals. I would like the authors to discuss these aspects for a more composite approach to this husbandry and to make this manuscript more comprehensive.  

I have a question for the authors to discuss - Is disbudding an accepted practice in organic livestock production as it is a routine procedure in intensive systems where spacing between animals is kept to a minimum? Disbudding is not natural and has welfare effects. Please clarify.

Response: We have added a statement that says that organic farmers still support disbudding, but they are apprehensive about pain mitigation methods.  They are also exploring polled genetics to alleviate the need for disbudding.

The manuscript is well written and nicely presented. However, the conclusions are very superficial and need to provide a broad overview and a take home message. At this moment I would like to see work on suggestions provided in my general and specific comments, before I recommend the manuscript for publication.

 Response:  Thank you for your thorough comments on the review.  They are very much appreciated.  We have added additional discussion on disbudding, mastitis, and dry cow therapy based on reviewer comments.  We agree that there is so much more than can be included on organic animal health, and that probably warrants under investigative literature review related to teat dips, milking machines, calves, vaccination, castration, AI, and nutrition.  This literature review was based on the first authors PhD literature review for her PhD degree and covered topics that she was researching throughout her graduate career.  We think we can include many other aspects in another literature review.  We have also rewritten the conclusions to provide a more broad overview.

Specific Comments

Line 10: Is this not the basis of every animal husbandry practice rather being exclusive for organic livestock farming? Please clarify or reframe.

Response: We agree.  We have removed the word “organic” from the sentence.

Line 21: Please replace ‘should be determined’ with ‘are expected’

Response: Agreed and changed.

Lines 22-24: Incomplete sentence. Please complete it.

Response: Agreed and we have rewritten the sentence.

Lines 33-39: How is this different from the intensive livestock farming? These problems are not exclusive for organic livestock farming. Please clarify.

Response: We Agree. These statements are not different from intensive farming and can be see on farms. We have removed the word “organic” from the sentence.

Lines 59-60: What about ‘intensive systems? Please clarify and comment.

Response: We have added words to the sentence to indicate that it also includes intensive farming systems.

Line 64: Please replace ‘unallowed’ with ‘not allowed’

Response: Agreed and changed.

Line 64-68: Please insert a reference to support this statement.

Response: We have included a reference. Thank you.

Line 81-84: Please insert a reference to support this statement.

Response: We have included a reference. Thank you.

Lines 89-90: I am sorry I am not able to understand the sentence. Please use a simpler language for the benefit of the reader.

Response: We have removed this sentence.  We decided that the statement was vague and did not really fit within the section.  Thank you for questioning this sentence.

Line 135: Please replace ‘consider’ with ‘deliberate upon’

Response: Agreed and changed.

Lines 381-430: These are challenges for the entire cattle farming industry whether organic, intensive, or conventional. Please discuss the dilemmas of organic farming in context of mastitis.

Response: We agree.  We have added some additional paragraphs in terms of mastitis and SCC on organic dairy farms. 

Lines 455-68: Please refer to studies on organic teat dips or organic dry cow therapies in this paragraph.

Response: We agree.  We have added some additional paragraphs in terms of organic dry cows on organic dairy farms. 

Line 488: Please replace ‘suspicious’ with ‘apprehensive’

Response: Agreed and changed.

Round 2

Reviewer 3 Report

Reviewer comments for manuscript ID animals-1772681entitled ‘Alternative practices in organic dairy production and effects on animal behavior, health, and welfare’ Round 2

General Comments

 I congratulate the authors for their hard work. I have a minor query that I have mentioned in the specific comments section. I am satisfied with the corrections done by the authors and recommend the publication of the manuscript.

Specific Comments

Line 275-79: How about using cornual nerve block for disbudding in calves? Please comment/clarify?

Author Response

Thank you for your generous comments.  

Line 275-79: How about using cornual nerve block for disbudding in calves? Please comment/clarify?

Response: We have added some sentences after Xylazine paragraph that indicates that Lidocaine is approved for organic production in the USA and have added some sentences describing lidocainte use as a cornual nerve block.